

# Baseline scenarios of heat-related ambulance transportations under climate change in Tokyo, Japan

Marie Fujimoto and Hiroshi Nishiura

School of Public Health, Kyoto University, Kyoto, Japan

## ABSTRACT

**Background**. Predictive scenarios of heatstroke over the long-term future have yet to be formulated. The purpose of the present study was to generate baseline scenarios of heat-related ambulance transportations using climate change scenario datasets in Tokyo, Japan.

**Methods**. Data on the number of heat-related ambulance transportations in Tokyo from 2015 to 2019 were examined, and the relationship between the risk of heat-related ambulance transportations and the daily maximum wet-bulb globe temperature (WBGT) was modeled using three simple dose–response models. To quantify the risk of heatstroke, future climatological variables were then retrieved to compute the WBGT up to the year 2100 from climate change scenarios (*i.e.*, RCP2.6, RCP4.5, and RCP8.5) using two scenario models. The predicted risk of heat-related ambulance transportations was embedded onto the future age-specific projected population.

**Results**. The proportion of the number of days with a WBGT above 28°C is predicted to increase every five years by 0.16% for RCP2.6, 0.31% for RCP4.5, and 0.68% for RCP8.5. In 2100, compared with 2000, the number of heat-related ambulance transportations is predicted to be more than three times greater among people aged 0–64 years and six times greater among people aged 65 years or older. The variance of the heatstroke risk becomes greater as the WBGT increases.

**Conclusions**. The increased risk of heatstroke for the long-term future was demonstrated using a simple statistical approach. Even with the RCP2.6 scenario, with the mildest impact of global warming, the risk of heatstroke is expected to increase. The future course of heatstroke predicted by our approach acts as a baseline for future studies.

## INTRODUCTION

Heatstroke is a heat-related illness in which a patient becomes unable to control his or her own body temperature because of continuous exposure to a warm environment (*Bouchama & Knochel, 2002*). In the United States, 7,000 patients were reported to have died of heatstroke from 1979 to 1997 (*Centers for Disease Control and Prevention, 2000*). In France, over 15,000 excess deaths are estimated to have arisen from the 2003 heat wave (*Vandentorren et al., 2006*). Another study in the United Kingdom documented 892 excess deaths in the summer that resulted from three heatwaves in 2019 (*Brimicombe et al., 2021*),

Corresponding author
Hiroshi Nishiura,
nishiurah@gmail.com

and called for a more inclusive approach to heat waves including an intensified mitigation policy against climate change. In Japan, heatstroke patients in the summer account for 1% of annual emergency transportations, and more than half of these are aged 65 years or older (*Fire & Disaster Management Agency (FDMA), 2022a*; *Fire & Disaster Management Agency (FDMA), 2022b*). As a "super-aged" nation, the country expects a continuing increase in the incidence of heatstroke, possibly with an exacerbation of prognosis because of aging within the elderly population.

In the area of climate change, physical studies of risks associated with natural disasters, including the highest temperature of the year (*Papalexiou et al., 2018*) and hydro-meteorological hazards (*Debele et al., 2019*), have been extensively conducted, but there have been relatively few studies of health-related outcomes. The small number of health-related studies include predictive analyses of the health effects—particularly mortality, as well as heat stress disorder—of uncomfortable temperatures; these have been conducted on a national scale (*Barnett, Tong & Clements, 2010*; *Fouillet et al., 2007*; *Kim et al., 2019*; *Lee, Röösli & Ragettli, 2021*; *Weinberger et al., 2019*), at a regional level (*Dimitriadou et al., 2022*; *Scovronick et al., 2018*), and on a global scale (*Guo et al., 2017*; *Mistry et al., 2022*). Heat-related health events are usually observed among elderly people, and therefore more effective countermeasures for protecting elderly people are required (*Rodrigues, Santana & Rocha, 2020*; *Ragettli et al., 2017*). *Sheffield et al. (2018)* showed that health effects of heat waves depend on age and race (or ethnicity), and occur even among children younger than five years old, who are considered as vulnerable as elderly people.

Predictive models of heatstroke have been studied with the eventual aim of preventing the disease and reducing the associated mortality. The wet-bulb globe temperature (WBGT) is known to be a useful environmental predictor of heatstroke events (*Budd, 2008*; *Ueno et al., 2021*). The unit of WBGT is the same as that of temperature. WBGT is divided into five levels to alert people to heatstroke, and action policies are provided for each level. The five levels are: less than 21 °C, 21 °C to 25 °C, 25 °C to 28 °C, 28 °C to 31 °C, and 31 °C or more (*Asayama, 2009*). In Japan, *The Ministry of the Environment (2022)* regularly communicates the risk of heatstroke, raising public awareness to promote prevention, and WBGT thresholds 28 °C and 31 °C are used for elevating the level of warning messages. When the WBGT exceeds 28 °C, the risk of developing heatstroke is said to increase, acting as a social trigger for the cancelation of events, for example, the cessation of exercise (*The American College of Sports Medicine, 2022*; *Japan Sports Association (JSPO), 2013*). In Japan, an epidemiological study was conducted to predict the prevalence of heatstroke among elderly patients in an indoor environment (*Kodera et al., 2019*). Moreover, machine learning has been employed to attempt to better predict such events over the course of time (*Ogata et al., 2021*; *Ikeda & Kusaka, 2021*). Nevertheless, predictive scenarios over the long-term future have yet to be formulated. Because of the availability of climate change forecast scenarios—such as those based on the Coupled Model Intercomparison Project Phase 6 (CMIP6)—climatological variables in high spatial and temporal resolutions are readily available for the long-term future (*e.g.*, up to the year 2100).

The purpose of the present study was to generate baseline scenarios of heat-related ambulance transportations in Tokyo, Japan using climate change scenario datasets. We

specifically explored Tokyo, because Tokyo is the prefecture with the highest number of heatstroke transportations per year. Severe heatstroke cases are predicted for the long-term future using climatological data, particularly the WBGT because it is easily computed from meteorological variables (*Ono & Tonouchi, 2014*). While it is known that computing WBGT is technically complicated, a simplified WBGT exists as an approximation, acting as a publicly-used indicator of heat stress along with environmental stress index (*Kong & Huber, 2022*). Baseline scenarios will be useful for understanding the possible future course of heatstroke, thereby providing a basis on which interventions, particularly adaptation policies, can be designed.

## MATERIALS & METHODS

### Epidemiological data

To construct a forecast model, we used heat-related ambulance transportation data for the period from 2015 to 2019. The latest two years, 2020 and 2021, were omitted to disregard the impact of coronavirus disease (COVID-19) on associated behaviors (*e.g.*, a stay-home policy was in place during the pandemic). Specifically, we used a dataset on the daily number of people transported by ambulance for heatstroke, which records every single transportation event and is openly published by the *Fire & Disaster Management Agency (FDMA) (2022a)*. The ambulance transportation case data were structured by age group; in the following analysis, we analyze the data by dividing the population into three age groups: 0–17 years, 18–64 years, and 65 years or older. In addition to case data, data on the daily maximum WBGT were obtained from the Ministry of the Environment (2022). In addition, the population size for Tokyo for the period from 2015 to 2019 in the abovementioned three age groups was obtained (*Tokyo Metropolitan Government, 2022*) to calculate the risk of heat-related ambulance transportations. For instance, in 2019, the population sizes of 0–17 years, 18–64 years, and 65 years or older were 1.86 million, 8.25 million and 3.08 million, respectively, and of these, 553, 2,322 and 3,171 heatstroke cases called for ambulance transportation.

The Intergovernmental Panel on Climate Change (IPCC) released its AR6 on August 9, 2021. They found that, by at least the late 2030s, global average temperatures are likely to have increased by 1.5 °C above pre-industrial levels (*IPCC, 2021*). In the present study, we obtained climate change scenarios based on CMIP6, published by the National Institute for Environmental Studies (*Ishizaki, 2021*). Of the five different scenario models available, we selected two: (i) Model for Interdisciplinary Research on Climate version 6 (MIROC6), which was cooperatively developed by a Japanese modeling community to precisely reflect the meteorological conditions in Japan, and (ii) Meteorological Research Institute Earth System Model version 2.0 (MRI-ESM-2.0), which was developed by the Japan Meteorological Agency and offers accurate forecasts of rainfall and humidity. MRI-ESM-2.0 was adopted because humidity plays a critical role in calculating the WBGT. For each model, three different time-series scenarios—Representative Concentration Pathway (RCP) 2.6, RCP4.5, and RCP8.5—with daily climatological data from 1900 to 2100 were obtained for each 1-km square of geographic mesh. Using the two models and

three time-series scenarios, the daily maximum temperature, relative humidity, total solar radiation, and average wind speed in a geographic space that contains the Imperial Palace of Japan, located in central Tokyo, were extracted to calculate the WBGT for each date (*Ono & Tonouchi, 2014*). The reason why we chose the area around the Imperial Palace as the location of our analysis is that data on the daily maximum WBGT were consistently available for this particular geographic space over the whole time period.

## Mathematical model

For the purpose of prediction, we exploited the relationship between the number of heat-related ambulance transportations per million population and the daily maximum WBGT. These heatstroke data were used to predict the per capita risk because we subsequently used the estimate multiplied by the predicted age-specific population size during the forecasting process. The model was fitted by age group (*i.e.*, 0–17 years, 18–64 years, and 65 years or older). Several different statistical models were employed to describe the dose–response relationship. The first is a classical model that captures a monotonic increase in the risk of heatstroke as a linear function of the WBGT after the WBGT exceeds a threshold value; this is referred to as the hockey stick regression method (*Yanagimoto & Yamamoto, 1979*). This classical model was specifically revisited because there is an existing notion of a threshold around a WBGT of 28 °C. Using the daily maximum WBGT ($T$), the number of heat-related ambulance transportations per million population $n(T)$ was modeled as

$$E(n(T)) = \begin{cases} \beta_1, & \text{for } T < T_w \\ \beta_1 + \beta_2(T - T_w), & \text{for } T_w \leq T \end{cases} \tag{1}$$

where $T_w$ is the WBGT threshold above which the risk of heatstroke abruptly increases. $\beta_1$ is the constant risk at a WBGT below $T_w$; $\beta_2$ is the gradient of risk when the WGBT exceeds $T_w$. The model was independently fitted to each age group, and $T_w$ was treated as a parameter because different age groups were expected to have different threshold levels (*i.e.,* different physical endurance levels).

The second model is an extension of the first. Frequently, the WBGT is expressed as one of five discrete levels, with the highest level being a WBGT of 31 °C or more; this implies another threshold that boosts the risk of heatstroke. With two threshold temperatures, $T_{w1}$ and $T_{w2}$, the expected number of transportations per million population $n(T)$ was modeled as

$$E(n(T)) = \begin{cases} \beta_1, & \text{for } T < T_{w1} \\ \beta_1 + \beta_2(T - T_{w1}), & \text{for } T_{w1} \leq T < T_{w2} \\ \beta_1 + \beta_2(T_{w2} - T_{w1}) + \beta_3(T - T_{w2}), & \text{for } T_{w2} \leq T \end{cases} \tag{2}$$

where $\beta_1$ is the constant risk at a WBGT below $T_{w1}$ and $\beta_2$ and $\beta_3$ are the gradients of risk when the WBGT exceeds 28 °C and 31 °C, respectively.

The third model is a phenomenological model that assumes an exponential increase in the risk of heatstroke. We allowed a threshold for this model as well. Let $r$ be the exponential rate of increase in risk and let $\beta_1$ be the constant risk below the WBGT threshold. We have

$$E(n(T)) = \begin{cases} \beta_1, & \text{for } T < T_w \\ \beta_1 exp(r(T - T_w)), & \text{for } T_w \leq T \end{cases}. \tag{3}$$

Assuming that the variations in the observed counts of heat-related ambulance transportations are sufficiently captured by a Poisson distribution with the expectation given by (1), (2), or (3), maximum likelihood estimation was performed to obtain optimal values of the parameters. The Akaike information criterion (AIC) was calculated to compare the model fit. The mean absolute error (MAE) of the prediction from the observed values was also calculated to evaluate the accuracy of prediction.

## Future prediction scenarios of heat-related ambulance transportations

We consistently used the following formula to calculate the WBGT from readily available climatological data (*Ono & Tonouchi, 2014*):

$$T_{WBGT} = 0.735 \times T_{asmax} + 0.0374 \times RH + 0.00292 \times T_{asmax} \times RH + 7.619 \times SR - 4.557 \times SR^2 - 0.0572 \times WS - 4.064 \tag{4}$$

which is most frequently used in Japan to estimate the WBGT. $T_{asmax}$ is the maximum daily temperature (°C), $RH$ is the relative humidity (%), $SR$ is global solar radiation (kW/m$^2$) measured by horizontally installed solar radiation meter , and $WS$ is the average wind speed (m/s). A validation study of this approximate equation has excellently shown that 98.3% to 99.8% of WBGT estimate involved a bias less than 1.0 °C. In Japan, WBGT is physically measured in 11 different locations (weather observatories), and an approximate WBGT using Eq. (4) has been officially adopted in other 829 observatory stations. The MIROC6 and MRI-ESM-2.0 meteorological data were substituted into (4) to calculate the daily WBGT values from 2000 to 2100. Because the bifurcation of temperature change begins in 2015 in the two climate change models, data from 2000 to 2014 were calculated using common data.

In each future year, the proportion of days with a WBGT value falling into each of five discrete ranges was calculated. This was calculated as the number of days with a WBGT value in each range divided by the total number of days (365). Because climatological variables fluctuate from year to year, the abovementioned proportions were calculated for every five years.

We examined the projected trends in the number of heat-related ambulance transportations, explicitly accounting for future demography. To do so, we first calculated the WBGT from May 1 to September 30 in each year, and used models (1), (2), or (3) to calculate the risk per million population. Even in the long-term future, we assumed that the heatstroke risk would occur only between May and September. The predicted risk was embedded onto the projected age-specific populations aged 0–14 years, 15–64 years, and 65 years or older in the period 2015–2100 (*The Climate Change Adaptation Information Platform, 2022*). To calculate the 95% confidence intervals of our predictions of heat-related ambulance transportations, the parametric bootstrapping method was employed, because parameters were inferred by maximum likelihood method and the variance–covariance matrix was reasonably computed. To account for parameter uncertainties of models (1), (2), or (3), 5,000 bootstrap resampling experiments were conducted and the 2.5th and 97.5th percentile points were taken.

### Data sharing statement

All datasets, including modeled climatological data for the period 2015–2100 and empirical data on heat-related ambulance transportations for the period 2015–2019 in Tokyo are openly shared as the online supporting material (Tables S1–S4).

### Ethical considerations

The present study used publicly available information. Because no private information was used, ethical approval was not required.

## RESULTS

Table 1 shows point estimates and 95% confidence intervals of the parameters for the three models. The models differed in the WBGT threshold value above which the risk of heatstroke begins to increase. It is worth noting that the estimated threshold temperature for elderly people (those aged 65 years or older) was lower than that for the other age groups in all three models. Of the three examined models, the exponential model yielded the smallest AIC for all age groups, and thus was deemed the best-fit model.

Figure 1 shows a comparison between the observed and predicted data for heat-related ambulance transportations, as predicted by using the WBGT. Both the dose–response phenomena of heatstroke incidence depending on the WBGT and variations in the risk depending on the WBGT should be noted. The observed data indicate that the risk of heatstroke became increasingly variable as the WBGT increased. In particular, there was a large variation in the risk of heatstroke in people aged 65 years or older, potentially reflecting the natural adaptation behaviors that may have been adopted when the WGBT became extremely high.

Figure 2 shows the distribution of MAE as a function of WBGT, by age group. In common with the abovementioned findings from Fig. 1, the MAE showed an increasing trend as the WBGT increased, and this phenomenon was observed for all three models that we examined. Using the hockey stick model, the maximum MAE was in people aged 65 years or older, with the value 31.1. Similarly, the maximum MAE with the two-step hockey stick model was 28.6 in the same age group. When the exponential model was employed, the maximum MAE was 28.1, again in people aged 65 years or older.

Figure 3 shows the time-dependent change in the proportion of high WBGT values in the long-term future, for each of the three climate change scenarios using two climate models (*i.e.,* MIROC6 and MRI-ESM-2.0). Using MIROC6, the number of days with a WBGT above 28 °C increased every five years by 0.16% for RCP2.6, 0.31% for RCP4.5, and 0.68% for RCP8.5. Similarly, using MRI-ESM-2.0, the number of days with a WBGT above 28 °C increased every five years by 0.20% for RCP2.6, 0.41% for RCP4.5, and 0.67% for RCP8.5. One noteworthy point is that the proportion of days with a WBGT above 31 °C monotonically increased in both models for all climate change scenarios.

Figure 4 shows the long-term forecast of heat-related ambulance transportations for the period 2000–2100 for each climate change scenario and model, using MIROC6. Even when the hockey stick model was employed with the RCP2.6 scenario (the combination that yielded the mildest impact of global warming), the total number of ambulance

Fujimoto and Nishiura (2022), *PeerJ*, DOI 10.7717/peerj.13838

**Table 1** **Parameter estimates and model fit of dose–response model for heat-related ambulance transport as a function of WBGT in Tokyo, 2015–19.** Each model was fitted independently to the data by age group. We used heat-related ambulance transported data from 2015–19 retrieved from the Fire and Disaster Management Agency share. $T_w$ represents threshold WBGT and its unit is Celsius degree. AIC stands for the Akaike Information Criterion, calculated assuming it follows a Poisson distribution. The hockey stick model assumes that the risk of developing heatstroke increases linearly when the WBGT exceeds the threshold $T_w$. Two-step hockey stick model assumes that the risk of developing heatstroke increases in a two-step manner, and the boundary thresholds are $T_{w1}$ and $T_{w2}$. The exponential formula assumes that the risk of developing heatstroke increases non-linearly when the WBGT exceeds the threshold $T_w$. CI stands for the confidence interval as calculated from bootstrapping method.

| | The hockey stick model | | | | Two-step hockey stick model | | | | | | Exponential model | | | |
| Age | AIC | $T_w$ | $\beta_1$ | $\beta_2$ | AIC | $T_{w1}$ | $T_{w2}$ | $\beta_1$ | $\beta_2$ | $\beta_3$ | AIC | $T_w$ | $r$ | $\beta_1$ |
|---|---|---|---|---|---|---|---|---|---|---|---|---|---|---|
| 0–17 years | 2493.8 | 28.8 | 0.6 | 1.8 | 2467.3 | 28.0 | 30.8 | 0.6 | 0.8 | 3.0 | 2456.0 | 25.7 | 0.4 | 0.5 |
| 95% CI | | (28.6–29.0) | (0.55–0.67) | (1.62–1.97) | | (27.6–28.4) | (30.4–31.1) | (0.53–0.64) | (0.58–1.04) | (2.40–3.54) | | (25.1–26.3) | (0.37–0.45) | (0.38–0.54) |
| 18–64 years | 1747.0 | 28.4 | 0.3 | 1.6 | 1650.8 | 26.1 | 30.6 | 0.2 | 0.5 | 3.1 | 1630.3 | 23.7 | 0.5 | 0.1 |
| 95% CI | | (28.2–28.5) | (0.26–0.34) | (1.49–1.76) | | (25.7–26.5) | (30.3–30.9) | (0.16–0.24) | (0.36–0.56) | (2.63–3.62) | | (22.8–24.6) | (0.44–0.48) | (0.09–0.17) |
| 65 years and older | 3999.9 | 27.9 | 1.0 | 4.4 | 3671.2 | 25.5 | 30.5 | 0.6 | 1.4 | 9.0 | 3620.2 | 22.7 | 0.4 | 0.4 |
| 95% CI | | (27.8–28.0) | (0.88–1.04) | (4.23–4.62) | | (25.3–25.7) | (30.3–30.7) | (0.52–0.67) | (1.22–1.50) | (8.16–9.79) | | (22.1–23.3) | (0.42–0.44) | (0.28–0.44) |

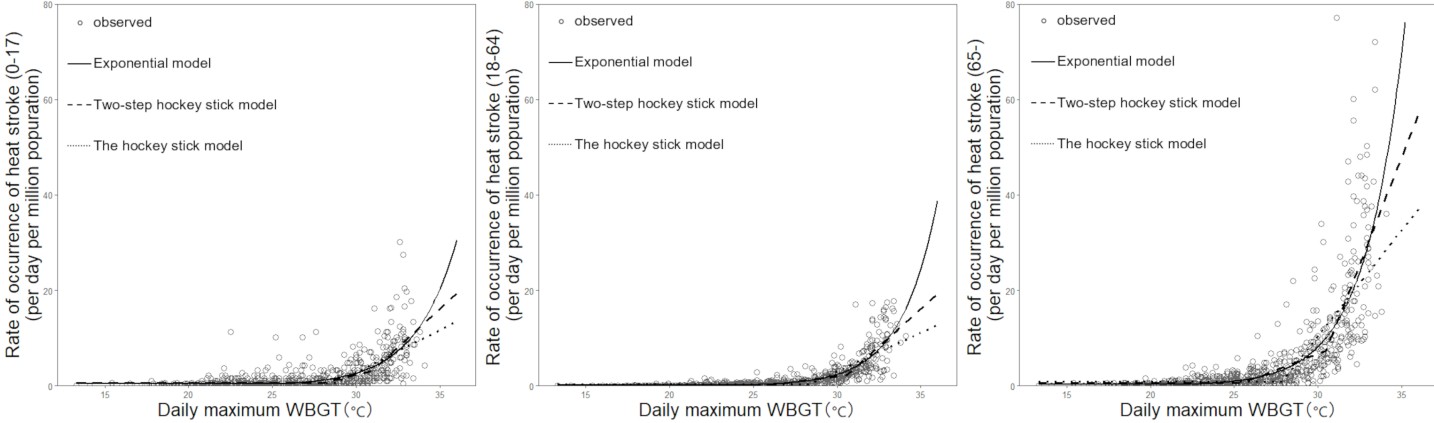

**Figure 1** **The risk of heat-related ambulance transportations as a function of wet-bulb globe temperature (WBGT): a comparison between observed and predicted data.** The vertical axis is the number of cases per million population per day, and the horizontal axis is the daily maximum WBGT. The left, middle, and right panels show the age groups 0–17 years, 18–64 years, and 65 years or older, respectively. White circles represent observed values, solid lines show the exponential model, long dashed lines show the two-step hockey stick model, and dotted lines show the one-step hockey stick model.

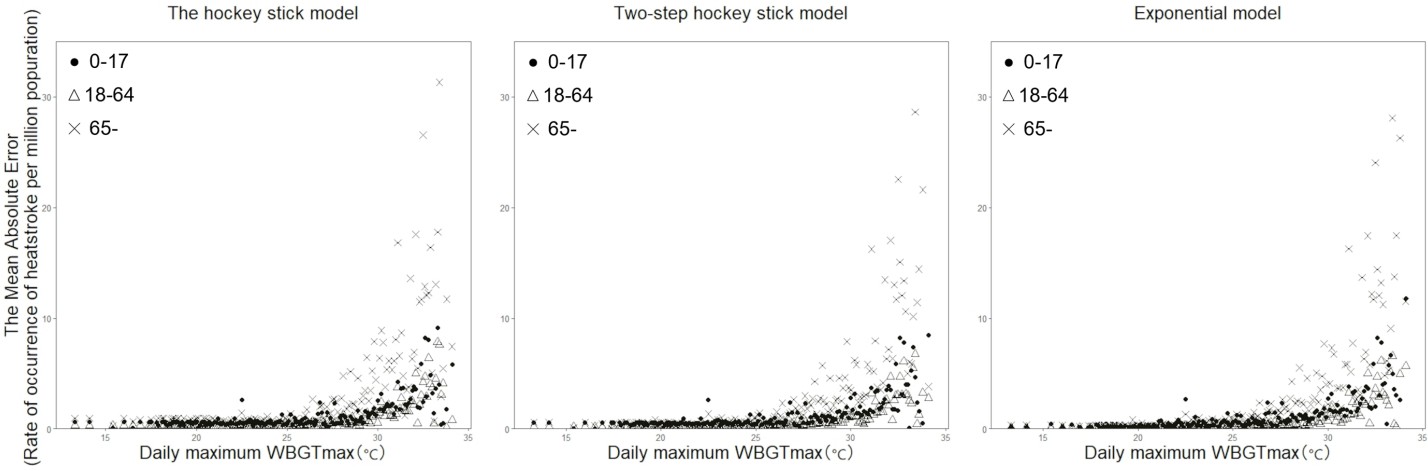

**Figure 2** **Mean absolute error between observed and predicted values, by age group.** The mean absolute error (MAE) between the observed and predicted data is shown for each model. Dots, white triangles, and crosses represent the MAE values for the age groups 0–17 years, 18–64 years, and 65 years or older, respectively. The MAE was calculated by dividing the absolute difference between the observed and predicted values by the number of data points for each identical wet-bulb globe temperature (WBGT) value.

transportations is predicted to increase. In 2100, the numbers of heatstroke patients transported in the groups aged 0–14 and 15–64 years are projected to be about three times higher than the numbers observed in 2000. Among people aged 65 years or older, the number of ambulance transportations is expected to be more than six times that observed in 2000. Figure 5 shows the same long-term forecast, based on MRI-ESM-2.0. In general, the forecast is qualitatively identical to that derived from MIROC6. The increase in risk

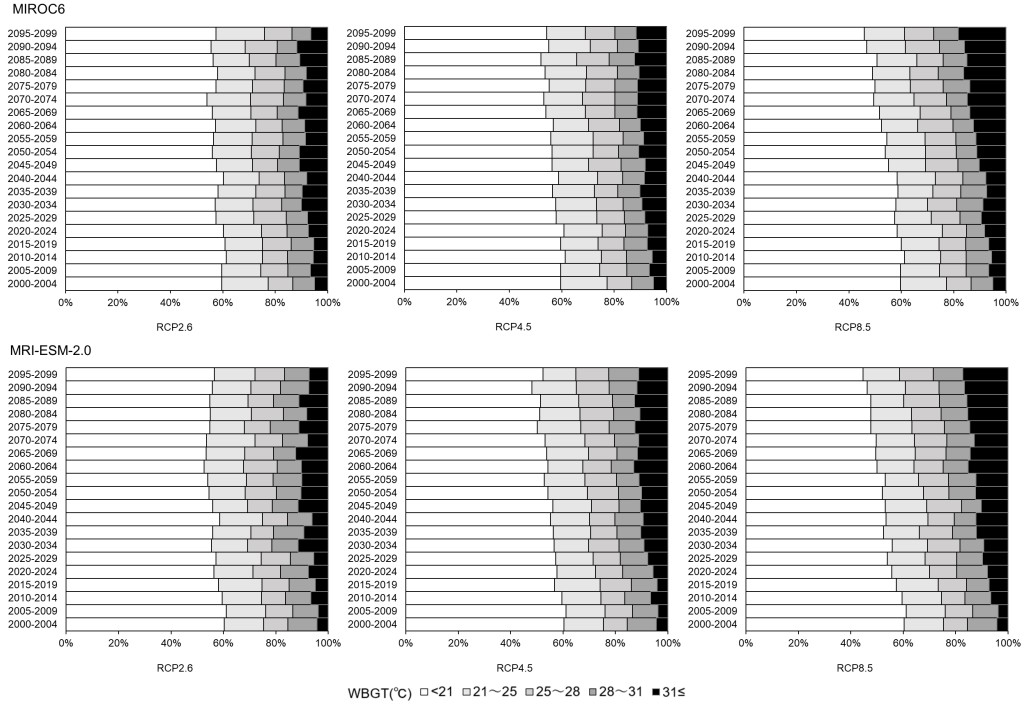

**Figure 3** **Future wet-bulb globe temperature (WBGT) patterns in Tokyo for the period 2000–2100.** WBGT is represented as five discrete groups. The top two groups (*i.e.,* a WBGT of 28–31 °C and a WGBT above 31 °C) are regarded as having a particularly high risk of causing heatstroke. The proportion of days that have each range of WBGT values was calculated for every period of five years. The top and bottom panels show the projections for the MIROC6 scenario and the MRI-ESM-2.0 scenario, respectively. The left, middle, and right columns show the predictions for the RCP2.6, RCP4.5, and RCP8.5 temperature increase scenarios, respectively.

over time among elderly people tends to be greater than that among younger groups in both models.

## DISCUSSION

The present study developed a model to forecast heat-related ambulance transportations for the long-term future until 2100 in Tokyo, Japan in various climate change scenarios and using multiple climatological scenario models. In addition to the climatological scenarios and models, three dose–response models were employed to predict the heat-related ambulance transportations per million population as a function of the daily maximum WBGT, using observations from 2015–19 as the training data. The elderly age group was revealed to have a lower WBGT threshold than younger age groups. Our finding is consistent with published studies which reported that the elderly age group tends to have a difficulty in regulating body temperature under warm environmental conditions (*Meade et al., 2020*; *Larose et al., 2013*). Calculating the WBGT for RCP2.6, RCP4.5, and RCP8.5 for the scenario models, we have shown that the proportion of days with a WBGT between 28 °C and 31 °C and the proportion with a WGBT above 31 °C will monotonically increase.

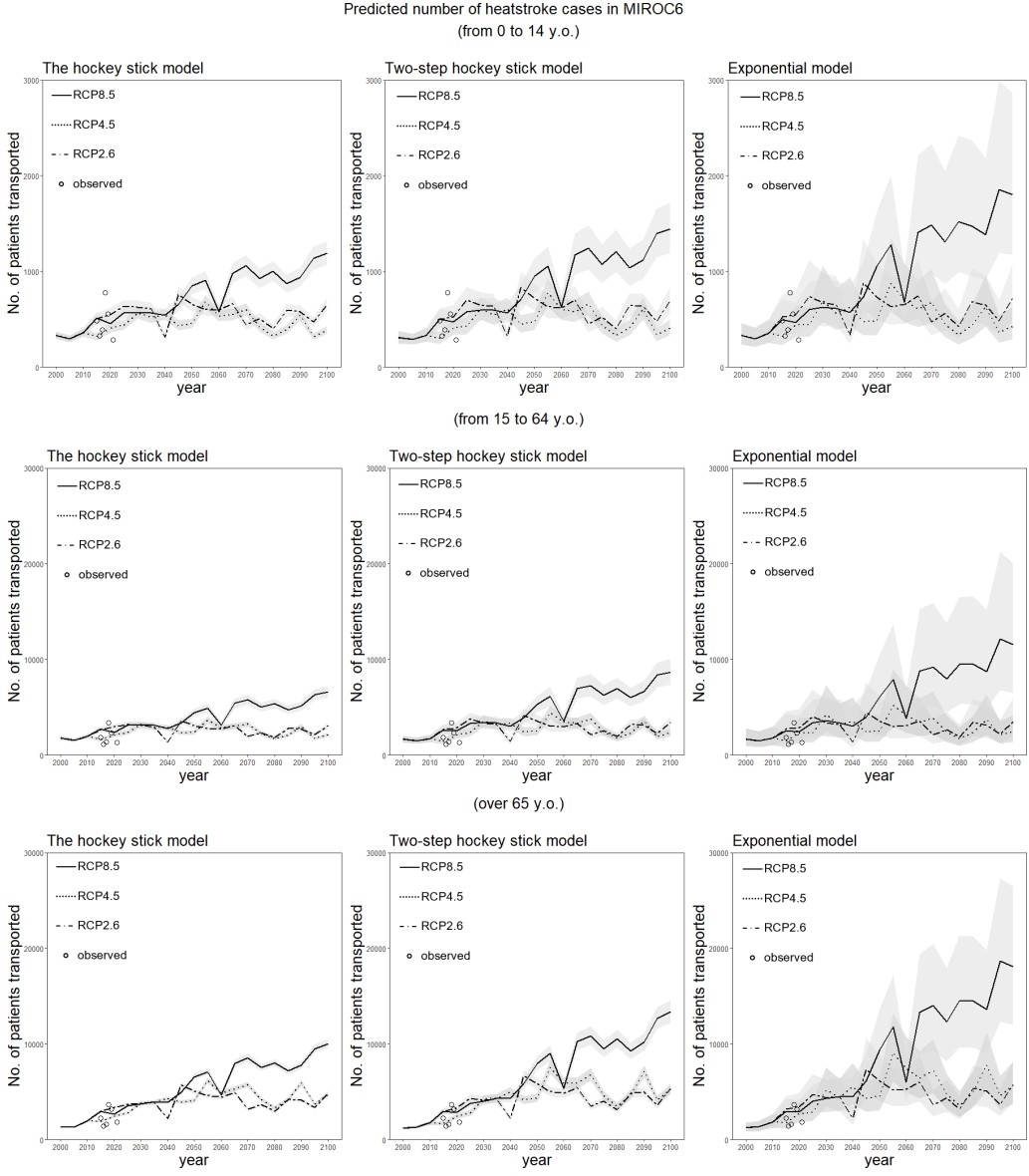

**Figure 4  Predicted number of heatstroke cases in Tokyo for the period 2000–2100, using the MIROC6 scenario.** The epidemiological prediction of the number of heat-related ambulance transportations for every period of five years is shown, using the daily maximum wet-bulb globe temperature (WBGT) derived from MIROC6. Predictions were made using the three statistical dose–response models. The 95% confidence intervals were calculated using the parametric bootstrapping method, and are represented by the shaded areas. The dotted and dashed line shows predictions with RCP2.6, the dotted line shows predictions with RCP4.5, and the solid line shows predictions with RCP8.5. The white circles represent observed data on heat-related ambulance transportations for the five-year period beginning in 2015.

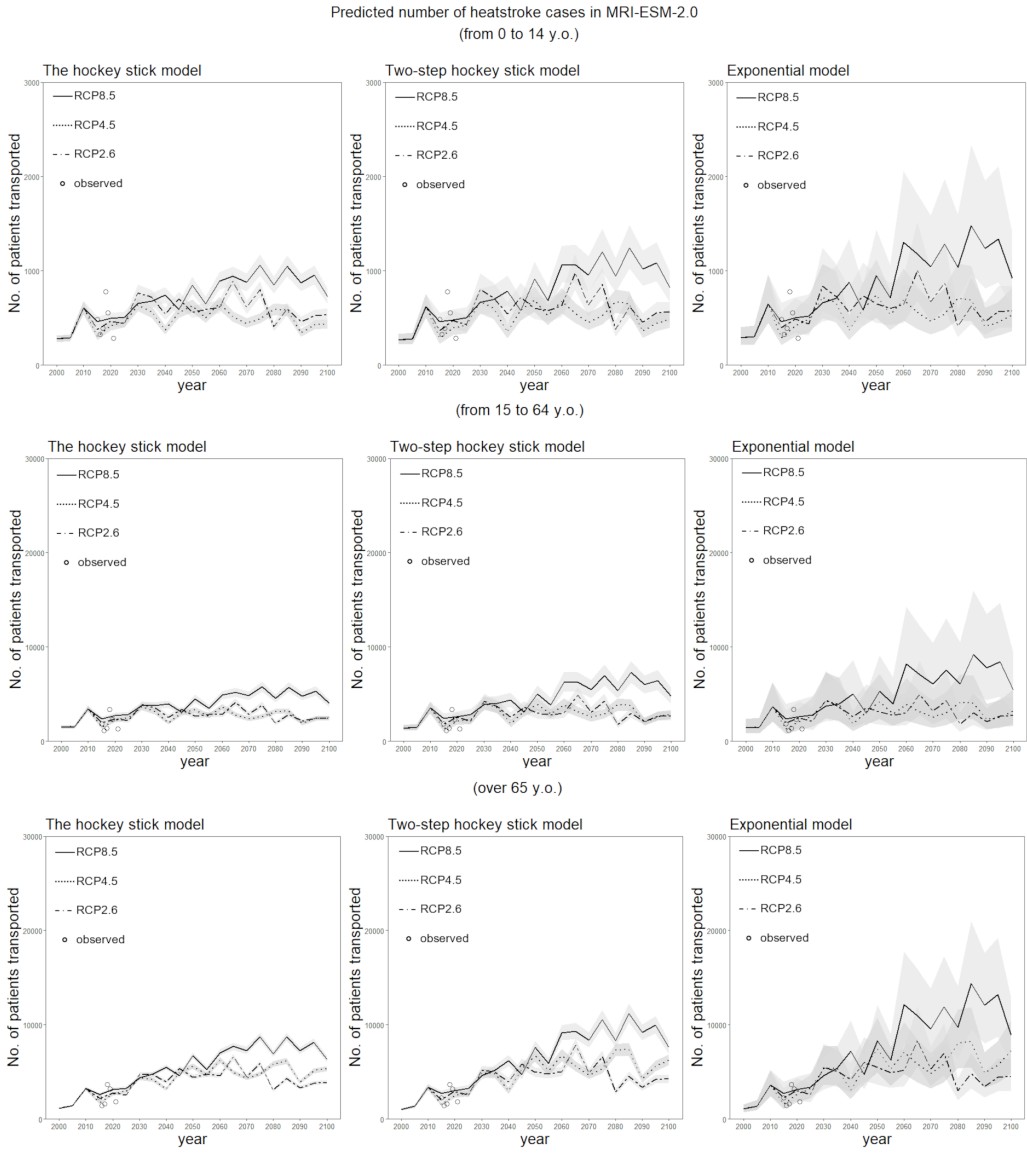

**Figure 5** **Predicted number of heatstroke cases in Tokyo for the period 2000–2100, using the MRI-ESM-2.0 scenario.** The epidemiological prediction of the number of heat-related ambulance transportations for every period of five years is shown, using the daily maximum wet-bulb globe temperature (WBGT) derived from MRI-ESM-2.0. Predictions were made using the three statistical dose–response models. The 95% confidence intervals were calculated using the parametric bootstrapping method, and are represented by the shaded areas. The dotted and dashed line shows predictions with RCP2.6, the dotted line shows predictions with RCP4.5, and the solid line shows predictions with RCP8.5. The white circles represent observed data on heat-related ambulance transportations for the five-year period beginning in 2015.

Embedding the predicted risk of heat-related ambulance transportations onto the future age-specific projected population, we were able to objectively predict the increased risk of heatstroke for the long-term future using a simple statistical approach. Even with the RCP2.6 scenario, which shows the mildest impact of global warming, the risk of heatstroke was predicted to increase.

To the best of our knowledge, the present study is the first to predict the risk of heatstroke using the proposed statistical dose–response modelling approach. The forecasting principles are simple and tractable (*e.g.*, using a dose–response model and embedding the risk onto the projected population), and yet the model captures the essential mechanisms of the elevated risk of heatstroke as a function of time. Only by quantifying the future risk of heatstroke, as practiced in the present study, can we understand the quantitative magnitude of the future increase. The future course of heatstroke predicted by our approach acts as a baseline for future studies. That is, various adaptation policies (*e.g.*, simply canceling outdoor events when the WBGT is extreme) will be planned according to the baseline risks, and we will examine how far the risk of heatstroke could be reduced by the adaptation efforts of society.

Of the three forecasting models that we examined, the exponential model yielded the smallest AIC value, indicating that the dose–response relationship between the risk of heatstroke and the WBGT is most appropriately captured by this model for all three age groups. This finding indicates that the elevation of the risk of heatstroke is accelerated as the WBGT increases. In the context of climate change, given that the higher the WBGT, the larger the error, it is statistically challenging to successfully quantify the risk of heatstroke with an extremely elevated WBGT in the future. The exponential model implies that the intrinsic risk of heatstroke with a very high WBGT is likely to be far greater than what we explored in this study using empirical data.

An important observation from the dose–response phenomena is that the variance of the risk increased as the WBGT increased. In fact, using the simple dose–response models, the MAE appeared to be large with high WBGT values. This variation was particularly noticeable among people aged 65 years or older. We believe that this variation was caused partly by natural adaptation behaviors. Because a high WBGT did not necessarily lead to large numbers of ambulance transportations of heatstroke patients, it is possible that dynamic changes in the WBGT may predict the risk of heatstroke better than the daily maximum WBGT. It has been speculated that the incidence of heatstroke is increased by an inability to take adaptation behaviors in each specific individual , such as participating in sports event or festivals among young individuals (*Kasai et al., 2017*) and continuously laying down in a warm room among elderly (*Abrahamson et al., 2009*). Also, the risk of heatstroke may be high when the WBGT suddenly increases. However, if a heat wave includes several consecutive days with high WBGT values, people may change their behavior to reduce the risk of heatstroke in the subsequent days. To capture the observed risk more accurately using available climatological variables is a subject for our future study.

As mentioned in the introduction, published studies using machine learning methods for heatstroke prediction modelling have already been widely recognized (*Ogata et al., 2021*; *Ikeda & Kusaka, 2021*). The aim of the present study was to generate a longer-term

baseline scenario for heatstroke emergency transportation in Tokyo, using a dataset of climate change scenarios and possibly employing a more simplistic and yet tractable approach. To accomplish the goal, a statistical dose–response modelling approach based on an approximate WBGT value was shown to reasonably predict the number of heatstroke cases in future Japan.

There are three limitations that need to be discussed. First, the present study examined only empirical data from Tokyo; thus, it is unclear whether its findings are applicable to a wider geographic area of Japan. Forecasting must be extended to geographic regions other than Tokyo. Second, the present study relied on ambulance transportation data, which represent only a small fraction of heatstroke cases. Therefore, patients with mild heatstroke were not counted. However, we believe that the overall time trends were captured even though only severe cases of heatstroke were counted. Third, we must revisit the reliability of the forecasting model in the future. Wide variations in the numbers of heat-related ambulance transportations when the WBGT value is high imply that our model cannot capture the peaks and troughs of heatstroke incidence during summer days with high WBGT values, particularly among people aged 65 years or older. There is scope for improvement in the accuracy of the models.

Although several key tasks remain, we believe that the present study successfully provided a baseline scenario for heatstroke in Tokyo for the future, until the year 2100, using a simple statistical modeling approach. Various adaptation policies will be planned according to the baseline risks, and we will examine how far the risk of heatstroke could be reduced by the adaptation efforts of society.

## CONCLUSIONS

In conclusion, we have shown that a simple dose–response model with WBGT as a predictor can reasonably quantify the future increase in the number of heat stroke cases. The present study developed a model to forecast heat-related ambulance transportations for the long-term future, until 2100 in Tokyo, Japan in various climate change scenarios and using multiple climatological scenario models. By calculating the WBGT for RCP2.6, RCP4.5, and RCP8.5 for the scenario models, we have shown that the proportion of days with a WBGT between 28 °C and 31 °C and the proportion with a WGBT above 31 °C will monotonically increase. Embedding the predicted risk of heat-related ambulance transportations onto the future age-specific projected population, we were able to objectively predict the increased risk of heatstroke for the long-term future using a simple statistical approach. Even with the RCP2.6 scenario, which shows the mildest impact of global warming, the risk of heatstroke was predicted to increase. The future course of heatstroke predicted by our approach acts as a baseline for future studies.

## ACKNOWLEDGEMENTS

We thank the Edanz Group for editing a draft of this manuscript.

### Funding

This study was supported by the Environment Research and Technology Development Fund (JPMEERF20S11804) of the Environmental Restoration and Conservation Agency of Japan. Hiroshi Nishiura received funding from Health and Labor Sciences Research Grants (20CA2024, 20HA2007, 21HB1002 and 21HA2016), the Japan Agency for Medical Research and Development (JP20fk0108140, JP20fk0108535, and JP21fk0108612), the JSPS KAKENHI (21H03198), and the Japan Science and Technology Agency SICORP program (JPMJSC20U3 and JPMJSC2105). There was no additional external funding received for this study. The funders had no role in study design, data collection and analysis, decision to publish, or preparation of the manuscript.

### Grant Disclosures

The following grant information was disclosed by the authors:
The Environment Research and Technology Development Fund of the Environmental Restoration and Conservation Agency of Japan: JPMEERF20S11804.
Health and Labor Sciences Research Grants: 20CA2024, 20HA2007, 21HB1002, 21HA2016.
The Japan Agency for Medical Research and Development: JP20fk0108140, JP20fk0108535, JP21fk0108612.
The JSPS KAKENHI: 21H03198.
The Japan Science and Technology Agency SICORP program: JPMJSC20U3, JPMJSC2105.

### Competing Interests

Hiroshi Nishiura is an Academic Editor of PeerJ.

### Author Contributions

- Marie Fujimoto performed the experiments, analyzed the data, prepared figures and/or tables, and approved the final draft.
- Hiroshi Nishiura conceived and designed the experiments, analyzed the data, authored or reviewed drafts of the article, and approved the final draft.

### Data Availability

   Raw data is available in the Supplemental Files.

### Supplemental Information

Supplemental information for this article can be found online at http://dx.doi.org/10.7717/peerj.13838#supplemental-information.

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
