# Peer review of "Baseline scenarios of heat-related ambulance transportations under climate change in Tokyo, Japan"

_PeerJ, doi:10.7717/peerj.13838_

## Round 0.1 · original submission · Minor Revisions

Please pay careful attention to the recommendations of both reviewers. In particular, please do ensure that your design and later discussion of your data is more clearly and robustly articulated.

Reviewer 1 ·

Basic reporting

Overall interesting results are presented that demonstrate extreme heat impacts on hospital transportation and how this is projected to change.
I would move the section about Tokyo starting from line 274 into the introduction as this is the region this study focuses on.
Line 49 - this figure quoted is for 3 heatwaves over the summer of 2019.

Line 88- the WBGT is not known for being easy to calculate (i.e. see Kong and Huber, 2021)

The conclusion could be restructured to better highlight the importance of these results and the wider importance expanded in the discussion.

Experimental design

WBGT is a useful metric for this type of study for many reasons some of which are touched upon in this manuscript, however it would be beneficial if the authors could comment on the accuracy of their choosen approximation (line 158).

Line 162 is Solar Radiation, could you provide more details on how this is obtained. Other WBGT approximations make use of total sky radiation proportion and downward solar radiation for example.

How and why were the WBGT thresholds used in this study choosen, and are they designed for this approximation of WBGT?

A simple bootstrapping method is used, could you add in more detail why this particular method was choosen.

Validity of the findings

Your findings are very interesting and well presented, to me I find the study similar to the one referenced in line 385 but with a different methodology which is something that could be changed on line 244.

As in section 1, there is scope for more discussion for example line 235 the elderly in ambulances lower than the overall population and the comment on degree days in line 269 to highlight 2 instances.

I also think focusing on what are the key questions your study answers moreso in the conclusion would be beneficial, especially because this is a very important study in terms of projecting heat impact.

Additional comments

I've recommended minor revisions, and would like to thank the authors for their important research on this area. I think just a little more clarification as I state above and a stronger conclusion is needed before publication.

Reviewer 2 ·

Basic reporting

The manuscript is well written with clear explanations.

Experimental design

The data that is used in the study are not clearly stated. Suggest stating the sample size, name of parameters, etc. The data can be supplemented as supplementary information.

Validity of the findings

Lack of comparing the findings with the literature/theory. Suggest to include discussion and comparison with other literature.

Additional comments

The study discussed more mathematical models rather than the concept of heat transfer and its effect.

---

## Round 0.2 · accepted · Accept

Thank for your patience with the review process. Both of the current reviewers have recommended that we accept your manuscript for publication.

Reviewer 1 ·

Basic reporting

no comment

Experimental design

no comment

Validity of the findings

no comment

Additional comments

I thank the authors for addressing my concerns and I have no further comments to make.

Reviewer 2 ·

Basic reporting

The manuscript has been revised by taking into account the feedback from reviewers.

Experimental design

All is good now.

Validity of the findings

Yes, this is performed.

Additional comments

The manuscript is revised and suitable to be published.